## REVIEW ARTICLE

# Linking graphene-based material physicochemical properties with molecular adsorption, structure and cell fate

Sachin Kumar[1] & Sapun H. Parekh [1,2]*

Graphene, an allotrope of carbon, consists of a single layer of carbon atoms with uniquely tuneable properties. As such, graphene-based materials (GBMs) have gained interest for tissue engineering applications. GBMs are often discussed in the context of how different physicochemical properties affect cell physiology, without explicitly considering the impact of adsorbed proteins. Establishing a relationship between graphene properties, adsorbed proteins, and cell response is necessary as these proteins provide the surface upon which cells attach and grow. This review highlights the molecular adsorption of proteins on different GBMs, protein structural changes, and the connection to cellular function.

Over the past decade, graphene and graphene-based materials (GBMs), such as graphene oxide (GO), reduced graphene oxide (RGO), and their chemical derivatives, have gained substantial interest for the development of biomaterials for tissue engineering applications. The unique physicochemical properties of graphene and GBMs have been shown to significantly influence cell response, as previously reviewed[1,2]. In our previous review article, we highlighted the influence of GBM physical (roughness, topography, conductivity, and lateral dimension), chemical (wettability, surface functional moieties, and chemical interaction), and mechanical properties on cell response, without explicitly discussing how GBMs influence protein or biomolecular interaction[3]. The effect of graphene physicochemistry on biomolecular interactions of proteins has been reviewed separately[4–6]. Since proteins on biomaterial surfaces are a key mediator of subsequent cell behavior for nanomaterials[7–9], the unique physicochemical properties of graphene offer exciting opportunities to control protein adsorption, orientation, conformation, and ultimately cell fate.

The adsorbed protein distribution and conformation on biomaterial surfaces provide the cell-interfacing topographical, physical, and biochemical cues for cells to interact with and respond to a material[10]. Studies have demonstrated that a material's ability to adsorb proteins (e.g., albumin, vitronectin (Vn), fibronectin (Fn), collagen, and laminin) from the serum of cell culture media plays a vital role in cellular attachment and function[8,11]. Moreover, adsorption of the same protein but with different conformations, which exposes different domains to cells, has been

[1] Department of Biomedical Engineering, University of Texas at Austin, 107 W. Dean Keeton Rd., Austin, TX 78712, USA. [2] Department of Molecular Spectroscopy, Max Planck Institute for Polymer Research, Ackermannweg 10, D-55128 Mainz, DE, USA. *email: sparekh@utexas.edu

shown to mediate integrin-based cell adhesion and trigger downstream signaling, directing cellular function and differentiation[12–16]. Thus, one can surmise that adsorbed proteins and other bioactive molecules from culture media influence cellular response in such a way that is tightly controlled by the underlying physicochemical properties of the biomaterial surface.

The initial protein-substrate interactions are critically important when designing biomaterial substrates, since they set the stage for how proteins will attach to the substrate surface. Surprisingly for GBMs—given their strong interest from the research community—a complete picture detailing (i) molecular conformation of proteins on different GBMs, (ii) how this influences cell–substrate interaction, and (iii) the connection to cellular response remains elusive. As GBMs are expected to become promising next-generation biomaterials for tissue engineering applications[17], there is a clear need to link the physicochemical interaction of the graphene surface with protein adsorption, protein structure, and cellular response. Developing such a multiscale understanding will involve integrating experimental, theoretical, and simulation studies.

In this review, we attempt to thread a line through the above-mentioned areas (i–iii) and present how graphene physicochemical properties can be tuned to control cell fate via the proteins coating the graphene surface. We hope this review will encourage more researchers to work in this burgeoning subfield, and provide quantitative experimental results with direct connection between the different areas (GBM–protein and protein–cell) that link cell function and GBM physicochemical properties.

## GBM production and cytocompatibilty

Since the discovery of graphene, numerous techniques and methods have been reported for production of graphene and GBMs[18,19]. Common methods for graphene synthesis include chemical vapor deposition (CVD) and mechanical exfoliation[18]. Other techniques for preparation of GBMs, such as GO (having enriched oxygenated functional groups like hydroxyls, carboxyls, and epoxies), RGO (having fewer oxygen-containing functional groups than GO, but still more than graphene), and their derivatives, include chemical exfoliation, chemical and thermal reduction, and chemical functionalization[20].

Application of graphene and graphene-derived materials in biomedical research has been limited due to latent cytotoxicity concerns[21]. Measuring the interaction of graphene and GBMs with numerous different cell lines and animal models to understand the mechanism of cytotoxicity is only now becoming routine[3,22]. Hydrophobic suspended graphene nanoparticles in aqueous media showed more toxicity than hydrophilic GO particles, due to rapid agglomeration of hydrophobic graphene covering the cell surface, purportedly limiting nutrient supply and inducing oxidative stress, causing cell death. Upon interaction, graphene sheets result in physical damage to the cell membrane[23]. Also, the uptake of nanoscale, non-functionalized graphene shows more cytotoxicity compared with large and functionalized graphene[23]. The biological response of graphene and GBMs—as reviewed below—must be taken in the context of potential complications with cytocompatibility, which depends on particle size, concentration, chemistry, processing methods, and purity of graphene substrates.

While cytotoxicity studies with GBMs and particles are growing, relatively few studies have shown how the interaction of graphene-based particles with proteins in the culture media can influence cell survival. In one example, graphene particles in suspension were shown to exhibit cytotoxicity due to direct interactions with the cell membrane, and the cytotoxicity effects of graphene and GO in suspension can be mitigated when pre-incubated with FBS. Pre-incubation in FBS was shown to form a thin protein coating on GBMs in suspension, limiting their direct interaction with cells, thereby minimizing cytotoxicity[24,25]. Bussy et al. demonstrated the effect of a GO suspension on different cell lines in serum-free HEPES-buffered salt solution (BSS), Dulbecco's phosphate-buffered saline (PBS), and Dulbecco's modified Eagle's medium with serum (DMEM-S). Cell membrane ruffling and shedding damage upon interaction with GO occurred in both BSS and PBS, but not in DMEM—presumably due to adsorption of proteins[26]. These results show that different parameters affect graphene-substrate cytotoxicity in both the absence and in the presence of proteins.

## GBM–biomolecular interactions

**Overview**. We begin by focusing on the protein–graphene interaction, and review example studies highlighting what is now known about this interaction. An obvious starting point is the interaction of serum proteins with different graphene-based surfaces, as serum is present in nearly all cellular experiments.

**GBM–serum protein interactions**. Carbon-based nanomaterials, especially graphene and its derivatives, have been shown to interact strongly with different serum proteins[4,27,28]. GO, a graphene derivative with rich, oxygen-containing functional groups on graphene, has been found to interact and adsorb many proteins found in serum-based culture media. GO, having negatively charged oxygenated functional groups at physiological pH, as well as the hexagonal aromatic graphene structure, promoted hydrogen bonding, electrostatic, hydrophobic van der Waals, and π–π interactions allowing it to interact with various proteins in serum. As a result, GO showed very high amounts of serum protein capture on its surface[27]. Chong et al.[29] demonstrated interaction of different serum proteins: bovine fibrinogen (BFG), immunoglobulin (Ig), transferrin (Tf), and bovine serum albumin (BSA) with GO and RGO[29]. Both GO and RGO showed serum protein adsorption in the following order: BFG > Ig > Tf > BSA (most to least) with GO showing higher adsorption compared with RGO. The difference in protein adsorption was attributed to differences in surface chemistry of GO and RGO, with GO offering a greater variety of different interactions compared with RGO; RGO would offer less hydrogen bonding and electrostatic interactions.

Consistent with the two graphene substrates having different available protein interaction mechanisms, adsorbed proteins were found to interact differently on GO and RGO surfaces. The presence of polar groups such as hydroxyls, carboxyls, and epoxides on GO promoted adsorption mainly through electrostatic interactions, whereas RGO protein adsorption was mediated primarily by van der Waals interactions[30]. BFG and Ig showed structural heterogeneity on the GO surface in comparison with RGO, which was attributed to the presence of multiple oxygenated moieties on GO and differences among native protein structures[29]. Interestingly, BSA and Tf showed structural rearrangement from α-helical to enhanced β-sheet conformation after adsorption on GO surface, as depicted by circular dichroism (CD) spectra in Fig. 1a. BSA was also shown to undergo unfolding on graphite surfaces as illustrated in Fig. 1b, suggesting that unfolding is common on aromatic carbon surfaces. During unfolding, BSA underwent conformational changes such that the lipid-binding domain of BSA moved toward the graphite surface due to hydrophobic interaction with graphite[31]. Chong et al.[29] found that proteins with exposed aromatic residues content like Trp, Tyr, and Phe residues first align with the graphene surface via π–π stacking as illustrated in Fig. 1c, similar to the interaction between pure aromatic amino acids and graphene[29]. The authors highlighted that solvent-exposed aromatic residues are involved in the initial protein adsorption, followed by later interaction of buried residues once the protein partially unfolds

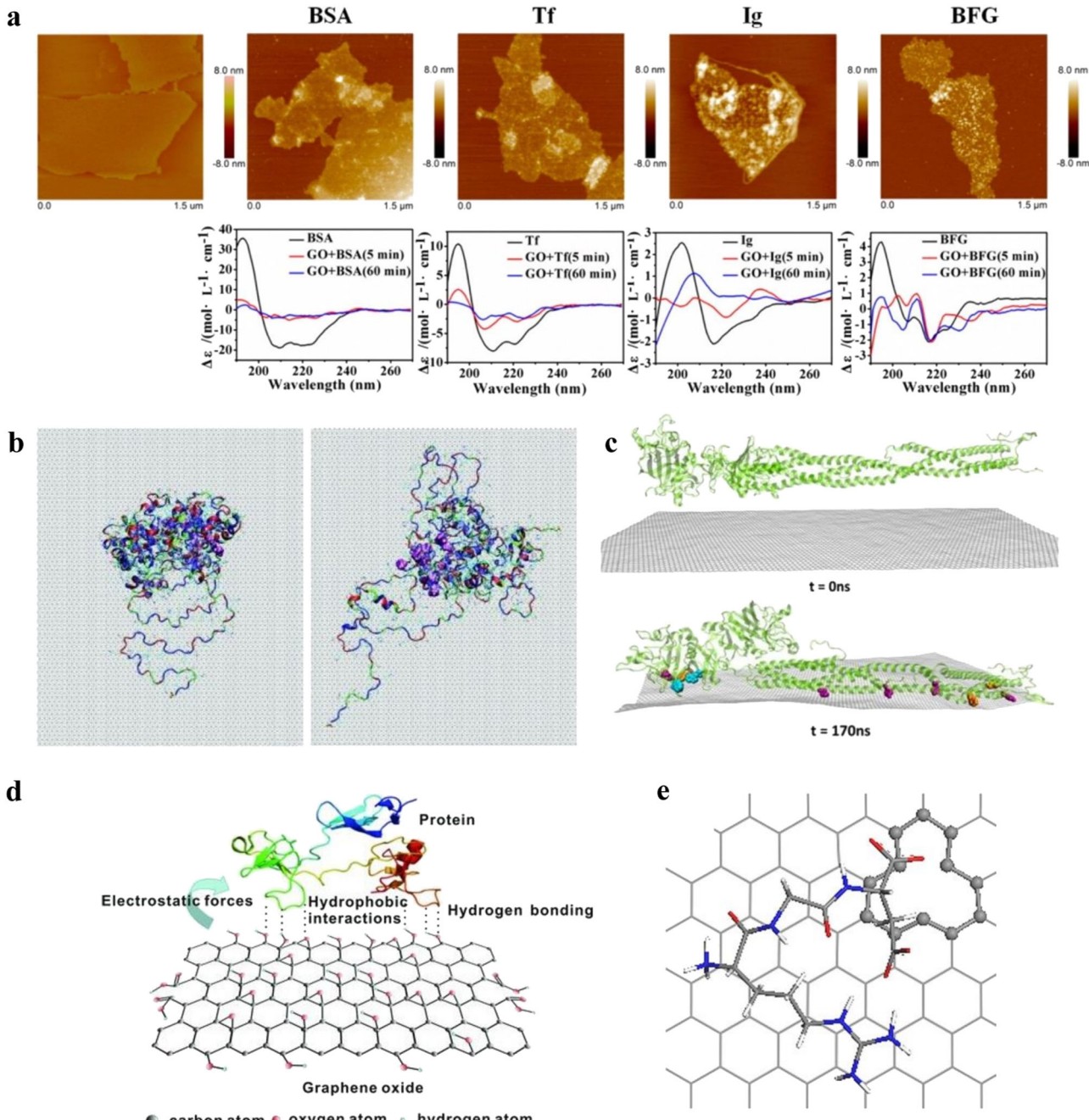

**Fig. 1 Multiple interactions between different serum proteins and graphene-based substrates mediate protein binding and conformation. a** Interactions between serum proteins BSA, Tf, IgG, and BFG, with GO along with the corresponding CD spectra, highlighting structural change with incubation time. CD spectra showed that BSA and Tf after adsorption on GO surface exhibited structural rearrangement from α-helical to enhanced β-sheet characteristics, whereas BFG and Ig showed structural heterogeneity on the GO surface (adapted with permission from ref. [29]. © 2015 American Chemical Society). **b** Simulation snapshot of BSA molecule after the 20-ns adsorption showing conformational changes with a decrease in α-helical content on a graphite surface (adapted with permission from ref. [31]. © 2011 American Chemical Society). **c** MD-simulated structural rearrangements of BFG on graphene for 170 ns. Protein aromatic residues Trp, Tyr, and Phe (highlighted in color) oriented and aligned with the graphene surface facilitate π–π stacking (adapted with permission from ref. [29]. © 2015 American Chemical Society). **d** Different interactions of serum proteins on partial oxidized graphene (adapted with permission from ref. [32]. © 2012 WILEY-VCH Verlag GmbH & Co. KGaA, Weinheim). **e** RGD is attracted to vacancy-defect graphene surfaces with mono-vacancy showing attraction to COO–. The vacancy is highlighted in ball-and-stick style (adapted with permission from ref. [34]. © 2015 American Chemical Society).

on the graphene surface[29]. Their results are suggestive of a positive correlation between protein adsorption and the amount of aromatic hydrophobic residues in a protein[29]. In another experimental study, Shi et al. showed how the reduction state of GO affected serum protein adsorption differently for different proteins[32]. In that study,

neat GO, partially reduced GO (pRGO) (having a low amount of oxygenated functional groups), and completely reduced GO (RGO) (lacking oxygenated functional groups) were prepared by thermal reduction, and adsorption of different proteins such as fibronectin and BSA was compared. Their results showed that pRGO and GO

exhibited better serum protein adsorption in comparison with RGO surfaces, consistent with Chong et al.[29]. Interestingly, pRGO showed the most adsorption of serum proteins, purportedly due to the presence of partial oxygen groups and partial hexagonal carbon structures. Oxygenated functional groups on pRGO surface introduced charged and electronegative regions for protein interaction through electrostatic and hydrogen bonding, while aromatic hexagonal carbon background of pRGO accommodated hydrophobic interactions with proteins as schematically shown in Fig. 1d. The independent studies from Shi et al. and Chong et al.[29] strongly suggest that strongest adsorption of serum proteins on pRGO was attributed to the mixture of electrostatic forces, hydrogen bonding, and hydrophobic interactions in pRGO with proteins[32].

Chemical defects, which are often created during functionalization and modification of graphene, are known to be "hot spots" for protein interaction. Structural defects on graphene typically alter the sp$^2$ carbon hybridization and distort the hexagonal benzene ring structure of graphene. This in turn affects protein binding due to steric hindrance and potential wrinkling of the graphene layer. Ebrahimi et al. used molecular dynamics (MD) simulations to study the role of graphene wrinkles and roughness on collagen interaction and adsorption. Their simulation results revealed that rough graphene surfaces show more adhesion of collagen fibers in comparison with smooth surfaces. As might be expected, the rough and wrinkled graphene surface provided more contact points between the graphene and collagen molecules for strong interaction[33]. Consistent with this finding, another theoretical study showed that the RGD cell adhesion peptide exhibited stronger interaction on vacancy defects in graphene (missing carbon atom in the lattice) in comparison with neat graphene[34]. The defects in graphene served as highly reactive sites for interaction between COO– of RGD peptide and vacancy defects on graphene as depicted in Fig. 1e. In another study, the presence of defects on a graphene surface was shown to increase interaction with fibrinogen through charge transfer, which further resulted in secondary structural changes in protein[35].

**GBM–hormone interactions**. In addition to serum proteins, graphene substrates have also shown strong propensity to interact with hormones present in serum[36,37]. A molecular simulation study demonstrated how insulin hormones present in serum can interact with graphene ribbon-like surfaces of different sizes[36]. The width of graphene ribbons showed direct correlation with insulin adsorption, and strong π–π interaction between phenyl rings in insulin and graphene surfaces was shown to promote stepwise conformational (secondary and tertiary structural) changes of insulin. On the other hand, Atabay et al. found that during adsorption of insulin on GO, insulin underwent configurational (conformation and rotational) rearrangement mediated by hydrophilic Ser, His, and Thr residues in chain B[37]. Interestingly, the anchored residues ultimately showed weak electrostatic interaction with the hydrophilic GO surface, which allowed the protein to rearrange after adsorption and restore its native structure. These studies again demonstrate the multiplicity (complexity) of ways in which even a single protein can interact with graphene, depending on the graphene surface chemistry.

**GBM–growth factor interactions**. Another important aspect of cell culture is use of specific growth factors or differentiation supplements in culture media. Interestingly, graphene-based substrates have also shown adsorption and interaction with growth factors and supplements present in culture media. Lee et al. reported that graphene showed more adsorption of osteogenic growth factor dexamethasone in comparison with GO[38]. The exceptionally high adsorption ability of graphene for dexamethasone can be again

attributed to π–π stacking between the aromatic rings in the dexamethasone and the graphene basal plane. Also we previously showed that polyethyleneimine (PEI)-functionalized GO (GO/PEI) showed high affinity to adsorb the osteogenic growth factor β-glycerolphosphate[39]. This was due to the presence of cationic PEI molecules on GO that provided electrostatic attraction to attract anionic phosphates like β-glycerolphosphate. In another study, GO's ability to adsorb a high amount of transforming growth factor-β3 (TGF-β3) was highlighted. GO having both graphene aromatic ring domains and oxygenated functional groups supported interaction with TGF-β3[40].

In summary, different serum proteins (hormones and growth factors) have complex interactions with graphene-based substrates. Depending on the physicochemical properties of graphene substrates, biomolecules interact and orient differently as highlighted collectively in Fig. 1. The presence of oxygenated functional groups on a graphene surface provides electrostatic and hydrogen bonding possibilities with proteins, while the aromatic benzene ring structure of neat graphene provides hydrophobic and π–π interaction to orient hydrophobic residues toward the graphene surface. Defects and surface nano wrinkles on graphene further influence protein interaction and morphology. Thus, it is very important to know and understand the specific surface properties of graphene substrate in order to appreciate the influence it has on protein interaction and conformation.

## GBM–cell interactions

**Overview**. Having reviewed how graphene physicochemical properties affect protein, hormone and growth factor adsorption, and structure, we move to the next level for evaluating graphene as a biomaterial, which is the graphene–cell interaction. Over the years, graphene-based particles have received substantial attention as future materials for tissue engineering applications, and authors have highlighted numerous physicochemical properties (surface topography, chemistry/wettability, and conductivity) of graphene substrates that influence, e.g., stem cell differentiation[3,41,42]. However, there is no consensus at this time about how different physicochemical properties of graphene substrates influence stem cell fate. The general paradigm for biomaterial–cell interaction is that, in the presence of media and serum, protein (and small molecules) adsorb and orient at the surface prior to cellular interaction[8]. It is virtually impossible to neglect the molecule (protein, small molecule, and hormone)–graphene interaction from the cell–graphene interaction. Ultimately, protein adsorption, conformation, and activity, in addition to other graphene physicochemical properties (surface topography, chemistry/wettability, and conductivity), control subsequent cell interaction[43,44]. Our goal in this section is to offer a perspective on cell response to graphene with a rationale starting from graphene–protein interaction on different graphene substrates and how the graphene-coating proteins affect cell behavior.

**GBM topography affects protein interaction and cell response**. Surface nanoroughness has been shown to play a critical role in protein adsorption and conformation[45]. Proteins show conformational (globular vs. elongated) change upon adsorption to nanoroughened surfaces where surface roughness is smaller, or larger, than dimension of protein molecule[10,46]. In addition, it was observed that the adsorbed protein monolayer surface topography/morphology was directly related to the surface topographical profile of the underlying material[47]. Thus, the adsorbed protein has been suggested to provide the ultimate topographical cues for cells.

Wrinkled graphene, which is essentially roughened graphene, was shown by Nayak et al. to have significant influence on stem cell response[48]. Surface roughness at the nanoscale translates into increased surface area for protein adsorption and orientation, providing increased number of cell-binding contact points[49,50]. On a similar note, Subbiah et al. showed that coating pure titanium (Ti) with GO provided nanoscale roughness for strong, homogeneous FN protein adsorption on the GO-coated Ti[51]. Subsequent biological studies showed more pre-osteoblasts attached, and more osteogenic differentiation features, on FN adsorbed to GO-coated Ti in comparison with pristine Ti or non-FN, GO-coated titanium (Tigra). The authors suggested that the availability of RGD motif of FN on GO surface caused better cell attachment and organization of the cytoskeleton. However, no direct measurements of FN molecular structure and conformation on the GO surface, nor information about cell-binding domain exposure, were presented in that work.

In another study by Hank et al., platelet aggregation in the presence of albumin and fibrinogen on hydrophilic GO and glass surfaces, with different roughness, was found to be distinct (Fig. 2a,b)[52]. GO showed high affinity for albumin and fibrinogen proteins on its surface in comparison with glass, as depicted in Fig. 2c. Remarkably, upon protein adsorption, the glass surface did not induce any conformational/structural change in proteins.

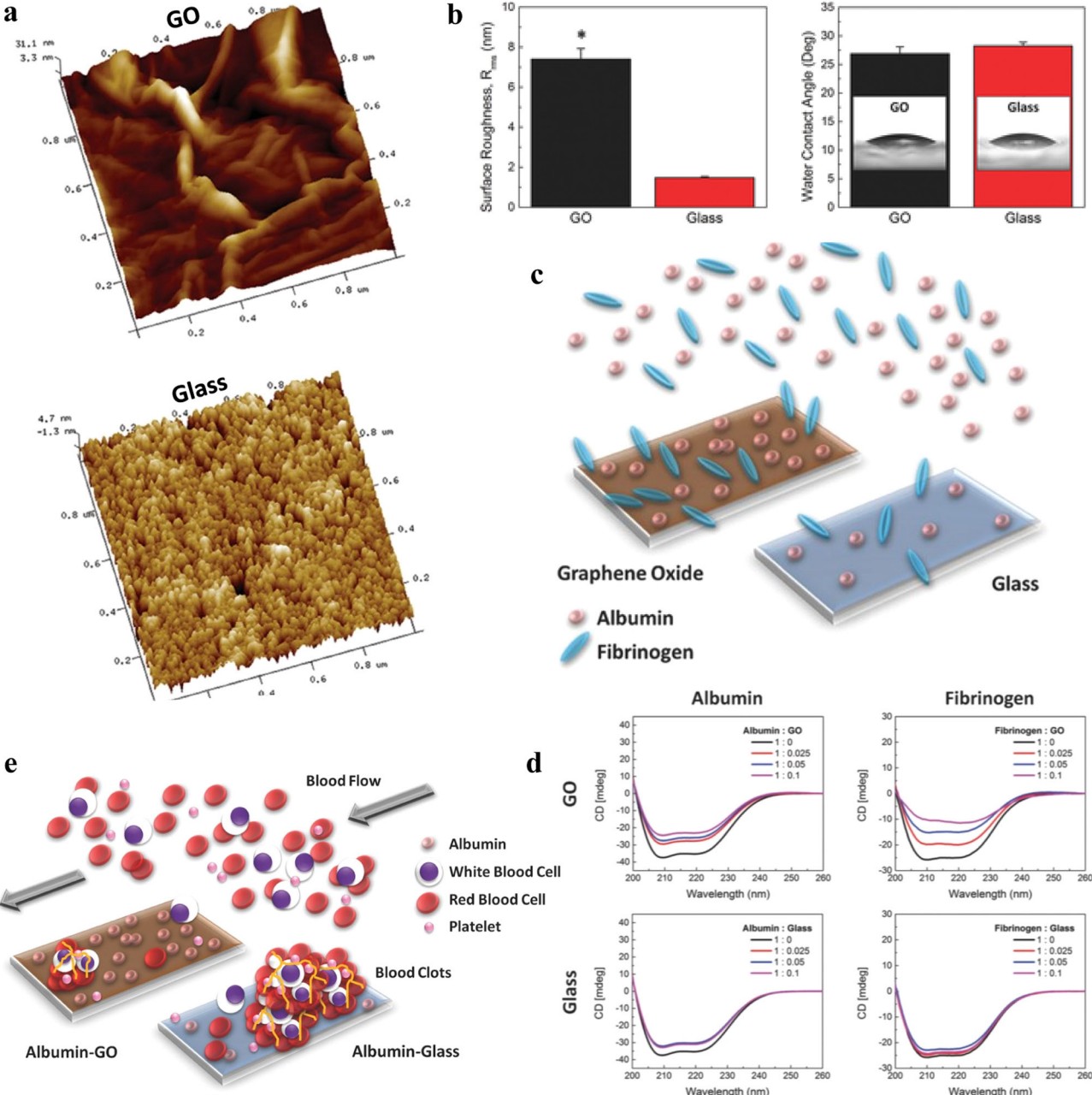

**Fig. 2 Graphene-substrate topography affects protein conformation and cell response. a** AFM 3D topographical images of GO and glass surface. **b** Difference in surface mean roughness and surface wettability with water contact angles of GO and glass. (The * indicates statistically significant differences for p-values < 0.05. The two-tailed Student's t test was used to make the pairwise comparisons). **c** Schematic illustrating more adsorption of albumin and fibrinogen on GO than on glass. **d** Circular dichroism (CD) spectra of albumin and fibrinogen incubated with GO and glass at different concentrations showing changes in protein structure for proteins incubated with GO. **e** Schematic illustrating the reduced formation of blood clots on albumin–GO surface compared with albumin–glass surface (adapted with permission from ref. [52]. © 2015 Wiley-VCH Verlag GmbH & Co. KGaA, Weinheim).

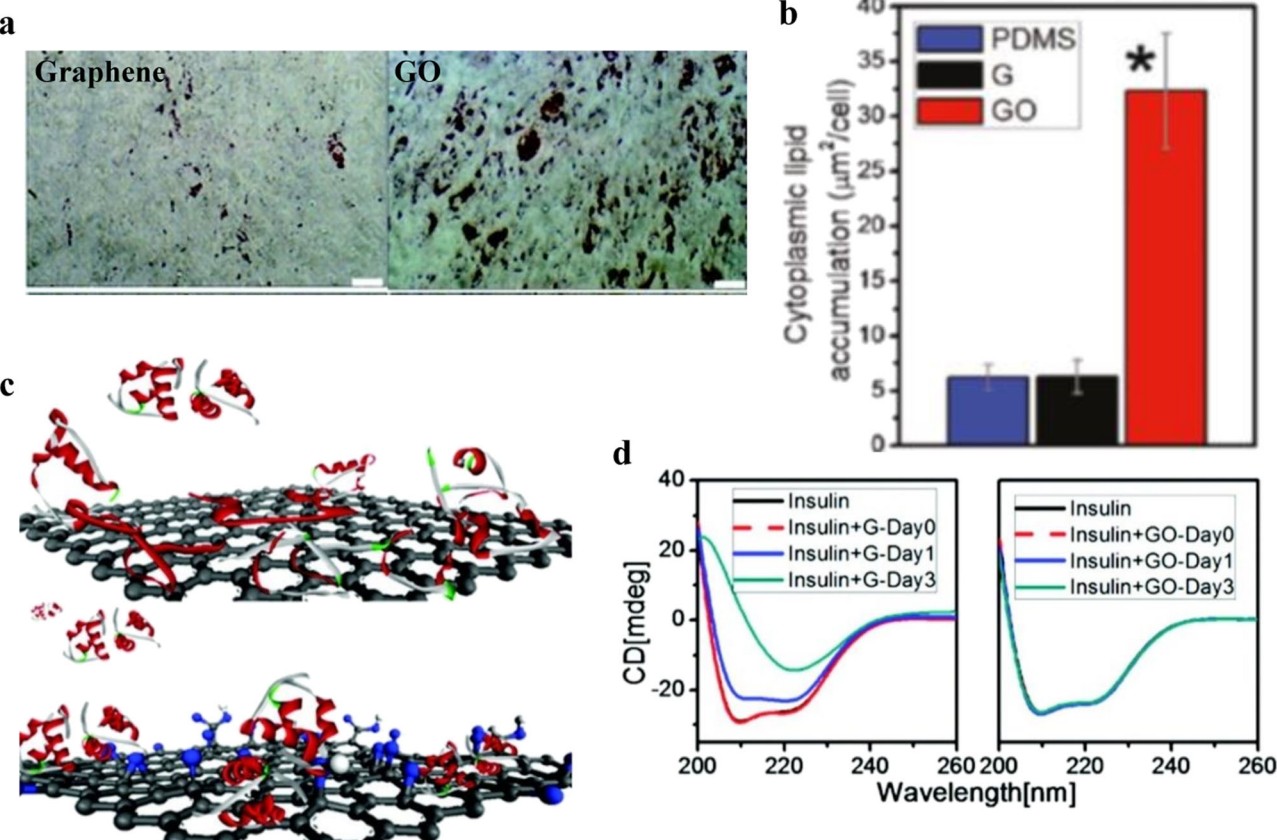

**Fig. 3 Graphene and GO substrate chemistry affects protein interaction and cell response. a** Cytoplasmic lipid accumulation assessed by Oil Red O staining after 14 days of induction on graphene and GO (scale bar 50 μm). (The asterisk indicates statistically significant differences for $p$-values < 0.05, using Student's $t$ test, $n = 4$ for each group). **b** Strong adipogenic differentiation of MSCs was observed on GO surfaces, with a significantly higher accumulation of lipid droplets. (The asterisk indicates statistically significant differences for $p$-values < 0.05, using Student's $t$ test, $n = 4$ for each group). **c** Schematic illustration of insulin adsorption on graphene and GO showing the respective conformations. Note that the schematics of the molecular substrate of protein (insulin), graphene, and GO are not scaling in proportion. **d** Far UV absorption CD spectra of insulin demonstrate the structural change upon adsorption on graphene and GO (adapted with permission from ref. [38]. © 2011 American Chemical Society).

However, GO induced secondary structural changes in both proteins as indicated by CD spectra in Fig. 2d. Albumin showed minimal conformational changes in α-helical secondary structures in comparison with large structural change in α-helix for fibrinogen. As a result, albumin retained its function as a surface passivator for platelet adhesion, as demonstrated in Fig. 2e. On the other hand, fibrinogen showed significant structural changes due to the α-helix domains unraveling, which was purported to prevent interaction between platelet surfaces through disruption of the αIIbβ3 integrin and fibrinogen[52]. This shows how graphene roughness can modify protein structure and downstream cell function.

**GBM chemistry influences protein interaction and cell response**. As described earlier, chemical modification or functionalization of graphene modifies surface chemistry and wettability of graphene-based substrates, which changes small molecule, protein, and ultimately cell interaction[53–55]. Lee et al. showed that graphene and GO substrates, with different surface chemistry and wettability, had different effects on differentiation of stem cells due to different interactions with adsorbed proteins and other biomolecules[38]. Their study demonstrated that GO promoted adipogenic differentiation of stem cells compared with graphene as shown in Fig. 3a,b. The ability of GO to influence adipogenic differentiation was attributed to high adsorption capacity of insulin, which mediates fatty acid synthesis and

adipogenesis[56]. High affinity and adsorption of insulin on GO surfaces was attributed to electrostatic interaction and hydrogen bonding with polar oxygenated functional groups on GO. After adsorption on a GO surface, insulin was shown to retain its three-dimensional conformation (Fig. 3c) and activity, thereby enhancing adipogenic differentiation. CD measurements supported these simulations, showing that the graphene surface altered the conformation of adsorbed insulin by reducing α-helix content, whereas the protein appeared to have the same structure on GO independent of adsorption time (Fig. 3d). Concurrent with this structural change, stem cells on graphene showed much weaker adipogenesis compared with GO. This work highlights the connection between graphene-substrate chemistry, interaction with insulin hormone, and a cellular response. It is tempting to connect the data and conclude that the substrate-induced structure of insulin caused more differentiation, but more work is needed to clarify this picture.

In another example by Depan et al., the authors nicely showed the interplay between protein adsorption (BSA in their case), protein morphology, and biological response on (hydrophilic) GO-modified chitosan[57]. In this study, they showed that biological function (cell attachment, proliferation (Fig. 4a), and mineralization (Fig. 4b)) of osteoblasts was enhanced on chitosan due to the presence of GO. Indeed, the presence of GO promoted more BSA adsorption in comparison with pristine chitosan—presumably due to hydrophilic groups on GO, as the topography

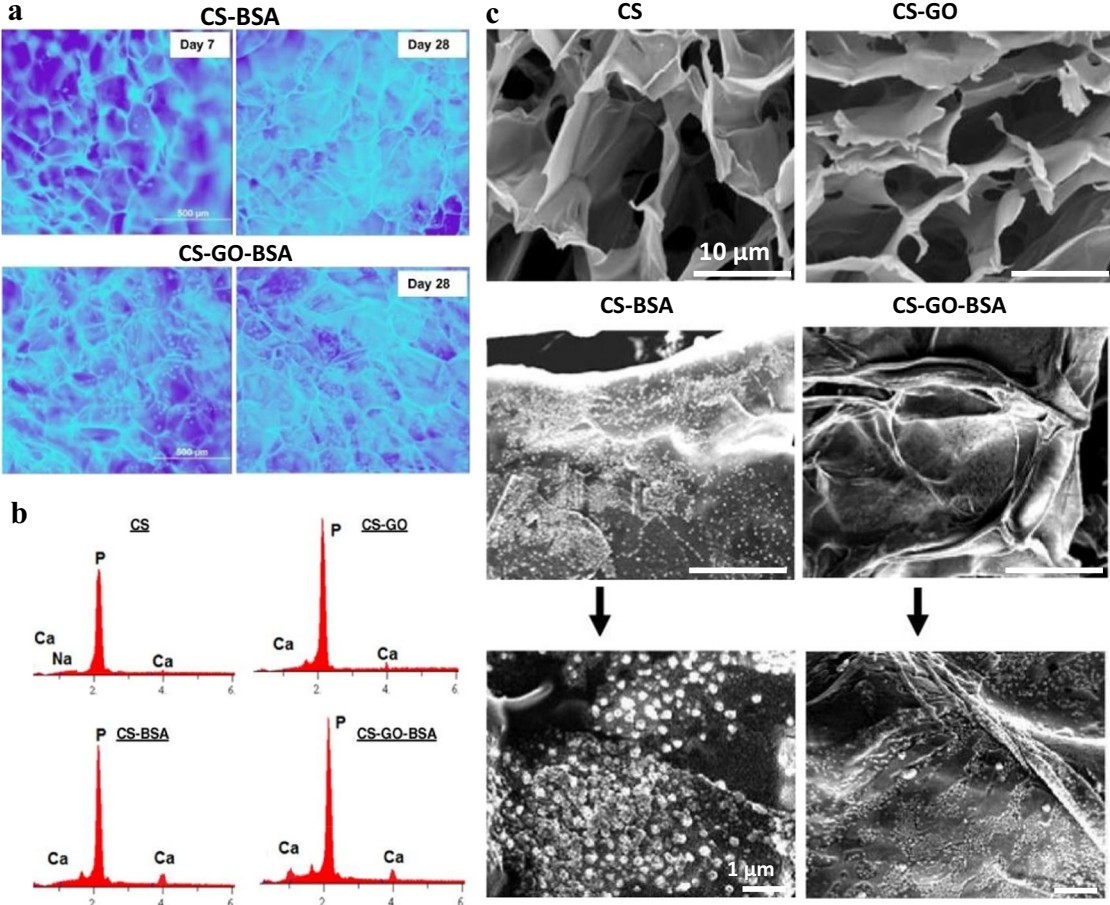

**Fig. 4 Graphene oxide chemistry affects protein morphology and cell response. a** Fluorescence micrographs illustrating pre-osteoblast proliferation on BSA-adsorbed scaffolds chitosan (CS–BSA) and chitosan–GO (CS–GO–BSA) after 7 and 28 days. Cells were stained with Hoechst to highlight nuclei, but due to autofluorescence of the scaffold nuclei appear as diffuse dots. **b** Osteogenic differentiation evaluated with energy-dispersive X-ray spectroscopy by mapping Ca and P mineral deposition on scaffolds at day 7. GO scaffolds with BSA adsorbed showed a higher amount of Ca and P mineral corresponding to higher osteogenic differentiation. **c** Scanning electron micrographs illustrating the structural morphology of pure CS; CS modified with GO (CS–GO) scaffold and morphology of adsorbed BSA protein on the respective scaffold (adapted with permission from ref. [57]. © 2012 Acta Materialia Inc. Published by Elsevier Ltd.).

of the two scaffolds was similar. Moreover, adsorbed BSA showed different morphology with regularly distributed smaller globules on GO-modified chitosan in comparison with randomly distributed large globules on neat chitosan as represented in Fig. 4c. Again, the authors suggested that subtle morphology differences in assembled BSA protein layer on GO-modified chitosan might be the reason for observed differences in cellular response[57]; however, the exact mechanism is unclear.

In a final example from this section, we highlight work by Ku et al. where they showed that graphene-based substrates were capable of influencing myogenic differentiation[58]. They cultured myoblasts on GO and RGO, and both showed elongated morphology in comparison with polygonal shape on glass. The authors suggested that elongated and cytoplasmic extension was caused by surface nanotopography provided by wrinkled structure of GO and RGO. However, when comparing RGO and GO surfaces, one sees that myoblasts on GO showed more myogenic differentiation by forming more multinucleated myotubes even though both had near-identical surface wrinkle roughness (Fig. 5a, b). Hence, one can surmise that the ability of cells to form more myotubes on GO than on RGO is heavily dependent on the differential chemistry of the surfaces in addition to the common nanotopography. When the GO and RGO substrates were incubated in differentiation media (DM),

GO showed significant increase in nitrogen content that can be attributed to enhanced protein adsorption (Fig. 5c). The authors correlated the enhanced myogenic differentiation of myoblasts on GO surface with high protein adsorption ability, particularly fibronectin (FN)[58]. Supporting this hypothesis, myoblasts cultured on polar surfaces (COOH and OH) showed strong binding of FN to cellular integrins and exhibited substantial differentiation into myotubes[59]. However, the FN conformation was not probed in these studies, so the connection to differentiation is still somewhat unclear[60–62]. The common undercurrent in all examples presented in this section is that the exact mechanism underlying all the examples shown here is ambiguous due to the complexity of the system.

**GBM conductivity influences protein interaction and cell response.** In addition to surface chemistry and topography of graphene, tissue engineering researchers have utilized graphene's surface conductivity/charge-carrying ability. Graphene's unique electrical and charge transfer properties have been reported to influence musculoskeletal and neuronal cell response[1,63]—both cells where membrane (electrical) polarization is fundamental for function. Several reports have described the interplay of electrical and chemical cues imposed by graphene to influence cell differentiation[64–66]. In one example, Li et al. showed that graphene

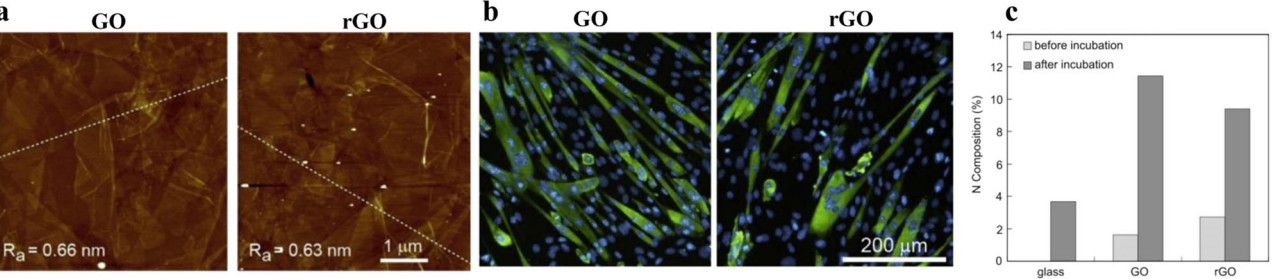

**Fig. 5 Graphene and RGO substrate chemistry affects protein adsorption and cell response. a** AFM micrograph of GO and rGO substrates and their respective average surface roughness (R). **b** Immunofluorescence staining for myosin heavy chain (MHC) showing more myotube formation on GO surface. **c** Change in nitrogen composition on the GO and rGO substrates before and after incubation in serum-containing media due to protein adsorption (adapted with permission from ref. [58]. © 2012 Elsevier Ltd.).

supported neuron growth by promoting better attachment and neurite sprouting of mouse hippocampal neurons when compared with standard polystyrene. The authors attributed this result to the complex interaction of graphene and chemicals in the culture medium in combination with electrical phenomena of graphene[67]. Feng et al., who investigated the effect of GO and RGO on neurogenic differentiation of the adipose-derived stem cells (ADSCs)[68], showed more effective differentiation of ADSCs into neuron-like cells on GO. This was despite GO having lower conductivity in comparison with RGO, encouraging the authors to suggest that chemistry "won" the competition between conductivity and chemical properties. The authors posited that the presence of oxygenated functional groups on GO surface influence protein adsorption, which mediated better interaction with cells to influence neuronal differentiation[69]. Thus, GO, with a rich amount of oxygenated functional groups and promoting protein adsorption in addition to the presence of more edge defects on GO, might have provided synergistic, localized electric fields for neuronal differentiation of ADSCs as illustrated in Fig. 6a. The mechanisms involved in these stimulatory behaviors, particularly regarding the electrical properties of graphene for neurogenesis, are still emerging.

In addition to the intrinsic conductivity of graphene, its charge, doping, and metal coating of graphene substrates can also be exploited to bias cell fate. Geng et al. revealed that coating germanium surfaces with graphene induced platelet adhesion and activation[70]. Coating graphene on the germanium surface was suggested to promote increased blood plasma protein (Fibrinogen) adsorption. Fibrinogen adsorption was further suggested to facilitate electron transfer from fibrinogen to graphene to germanium. Electron loss from fibrinogen has been shown to unfold and lead to formation of the fibrinopeptide and fibrin monomer, competent for polymerization[70,71], and fibrin is known to have a strong interaction with platelets through the $\alpha_{III}\beta_{II}$b integrin[72]. Thus, graphene acting as an electron acceptor and transporter, resulting in electron extraction from fibrinogen, could lead to enhanced platelet activation as illustrated in Fig. 6b. Because tissue regeneration requires a broad spectrum of bioactive macromolecules[73] and stimuli to support different types of cell growth, harnessing the combined effects of electrical and chemical properties of graphene substrates is very attractive[74–76].

**GBM interaction with growth factor influences cell response.** To conclude this section, we would also like to highlight how different graphene-substrate–growth factor interactions can act to pre-concentrate various growth factors and differentiation chemicals in order to influence stem cell response. One key aspect of differentiating stem cells to specific lineages is using specific growth factors, or differentiation supplements, in culture media.

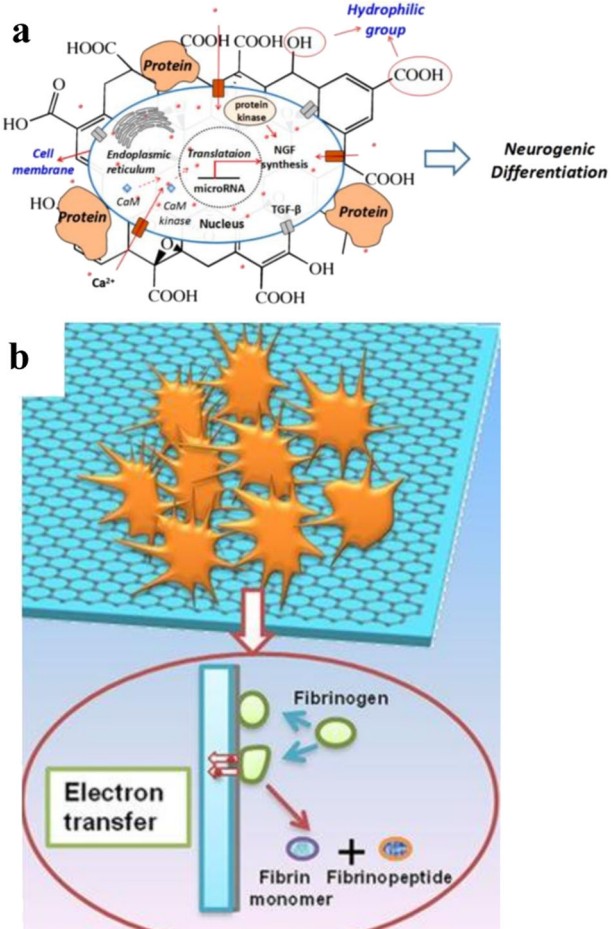

**Fig. 6 Graphene-based substrate electrical properties affect protein adsorption and cell response. a** Schematic illustration of protein adsorption from culture media on GO and possible localized electric field mediating calcium-dependent pathway for neurogenic differentiation of ADSCs (adapted with permission from ref. [68]. © 2018 Elsevier B.V.). **b** Activation of platelet by fibrinogen-adsorbed graphene mediated through electron transfer (adapted with permission from ref. [70]. © 2016 Springer Nature) (Note: schematics of protein, GO, and graphene are not scaling in proportion).

Adsorbed growth factors/differentiation supplements on graphene-based surfaces are reported to further influence the biological outcome of stem cells[38,39]. Lee et al. reported that graphene showed more osteogenic differentiation in comparison

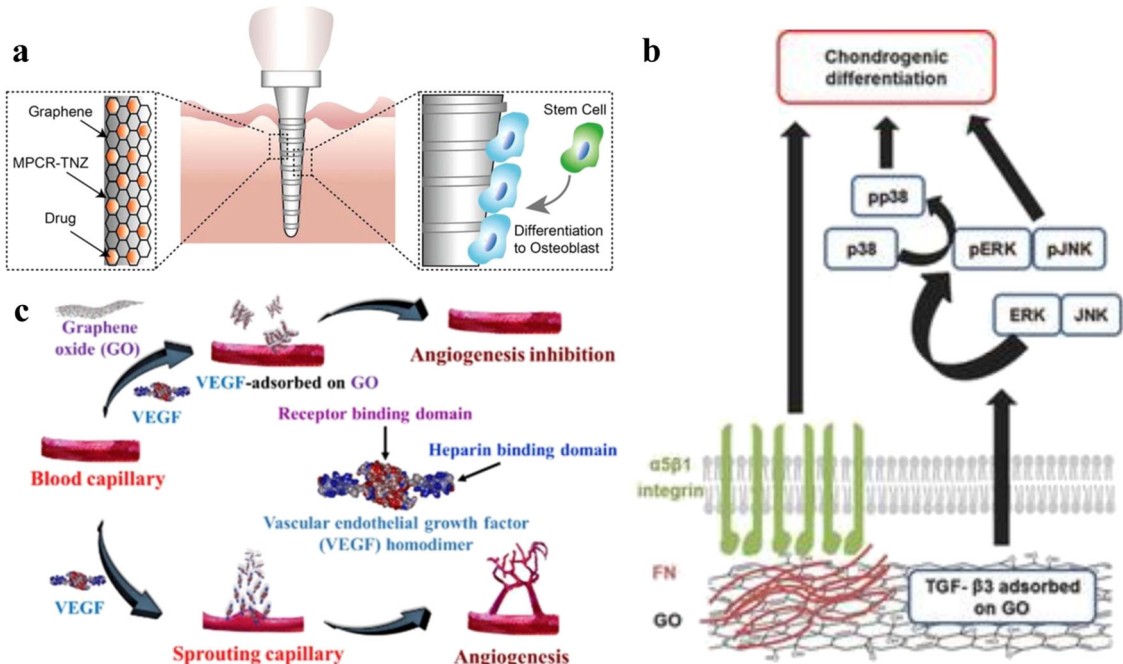

**Fig. 7 Growth factor–graphene-based substrate interaction. a** Schematic illustration for multi-pass caliber-rolled (MPCR) Ti alloy surface coated loaded with RGO adsorbing dexamethasone and promoting osteogenic differentiation of the stem cell for dental application (adapted with permission from ref. [77]. © 2015 American Chemical Society). **b** Schematic showing the underlying mechanisms of cell interaction with adsorbed fibronectin and response of TGF-β3 growth factor influencing cell signaling to enhance chondrogenic differentiation (adapted with permission from ref. [40]. © 2014 WILEY-VCH Verlag GmbH & Co. KGaA, Weinheim). **c** Diagram showing how adsorbed VEGF on GO inhibited angiogenesis due to the structural change in VEFG upon interaction with GO (adapted with permission from ref. [78]. © 2016 Elsevier Ltd.).

with GO due to high adsorption of osteogenic growth factors such as dexamethasone and β-glycerolphosphate[38]. Similarly Jung et al. exploited the ability of RGO-coated Ti alloy to interact with osteogenic dexamethasone (Dex) via π−π stacking on the graphitic domains of RGO to enhance osteogenic differentiation (Fig. 7a)[77]. In another study, GO's ability to adsorb a high amount of transforming growth factor-β3 (TGF-β3) was exploited to promote chondrogenic differentiation of adult stem cells. TGF-β3 retained its native structure on a GO surface for interaction with TGF-β receptors on a stem cell surface, and significantly influenced stem cell adhesion and chondrogenic differentiation as illustrated in Fig. 7b[40].

Growth factor adsorption has also been shown to promote undesired biological outcomes. For instance, Lai et al. showed that GO can efficiently bind a vascular endothelial growth factor (VEGF), depending on the oxidation state[78]. The high degree of oxidation of GO promoted more VEFG binding, and the highly basic surface charge of the heparin-binding domain of VEFG played a significant role in binding to GO surfaces through electrostatic and hydrogen interactions. Upon binding, VEFG showed structural changes with a decrease in secondary β-structure and increase in random coil structure. The authors went on to show that strong binding of VEFG by GO from plasma hindered the interaction with VEGF cellular receptors, inhibiting the proliferation, migration, and tube formation of human umbilical vein endothelial cells as illustrated in Fig. 7c.

## Outlook

Biomaterials influence cell function primarily through cell-material surface interactions where the identity and structure of adsorbed molecules, namely proteins, hormones, and growth factors, strongly determine the bioactivity of the material in context. The underlying physicochemical properties of the biomaterial ultimately regulate the entire cascade of interactions from molecular adsorption and structure to what cells ultimately "feel". Graphene, as an emerging material with unique physico-chemical properties, has been shown to stimulate different cell response and protein interaction due to the multiplicity of its interaction pathways. The potential of graphene for biomaterials is enormous because the biomaterial community can exploit the unique topographical, chemical, and electrical properties of the material for specific purposes. In this review, we try to highlight the diversity of graphene's properties and biomolecular interactions to emphasize the opportunities, and complexity, of using graphene substrates as biomaterials. Specifically, because of graphene's unique nature, harnessing its power requires understanding what is adsorbed, how adsorbed molecules look (structurally), and the (specific) cell response.

Going forward, the synergetic effort of experimental, theoretical, and simulation studies will be needed to develop methodologies and rational models, which allow connecting the substrate–protein interaction to cell function on different graphene substrates. Use of advanced analytical instruments like nano IR-AFM may provide protein topography and conformation in combination with IR spectroscopic information of graphene substrates at nanoscale. This information, along with the ultimate cell response, may provide the functional coupling of topography and molecular structure of the adsorbed proteins with cell function on graphene substrates having different physico-chemical properties. Finally, additional research is needed to evaluate cellular response on different graphene surfaces with well-defined biochemical environments and serum-free media formulations to reduce interference from these proteins, so that a clear conceptual picture can be drawn. In writing this review, we often found that unequivocal statements were hard to find due to the multiple ways graphene can affect protein structure, and how protein structure affects cell fate. Overall there is much to be done in the graphene–biomolecular interaction–cell response

context that can help provide design criteria for graphene bio-materials for specific needs.

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

## Acknowledgements

S.K. acknowledges Alexander von Humboldt Foundation Postdoctoral Fellowship, and S.H.P. acknowledges support from the Welch Foundation (F-2008-20190330) and Human Frontier in Science Program (RGP0045/2018). We thank Anika Keswani for helping to edit this paper.

## Author contributions

S.K. conducted the literature search and discussed literature examples with S.H.P. S.K. and S.H.P wrote and edited the paper.

## Competing interests

The authors declare no competing interests.
