## [Peer review file · Communications Chemistry]

REVIEWERS' COMMENTS:

Reviewer #1 (Remarks to the Author):

Recommendation: Reject

Comments to author

Kumar and Parekh reported on the review of "Controlling cell fate 1 with graphene physicochemical properties: a rich, multiscale challenge". The authors covered the manuscript including a. highlighting initial molecular confirmation of proteins on different physicochemical graphene-based surfaces, b. protein conformational changes, c. influence on cell-substrate interaction, and d. the connection to cellular function. Already, a series of reviews on graphene based materials have been reported previous. Almost similar type of review entitled "Interactions of graphene with mammalian cells: Molecular mechanisms and biomedical insights" was reported in *Advanced Drug Delivery Reviews* 105 (2016) 145–162. Therefore, this review does not have any significant novelty. In addition to this, some issues are associated with the manuscript which are described below:

1. The introduction section should be more informative.
 2. As the synthesis of graphene or graphene based materials is very important, so it should be included in the review.
 3. The authors should give a scheme which describes the overall idea of the review.
 4. The toxicity of graphene based materials has a major issue in biomedical application. Therefore, the toxicity effect should describe elaborately.
 5. The title describes that the authors have covered the interaction between the graphene and different types of cells but the authors mainly focused on tissue engineering related work. If they want to describe on tissue engineering, then the title should be changed accordingly.
- There are lots of issues throughout the manuscript. Therefore, the manuscript is not recommended for publication in this reputed journal. The manuscript should go more specified journal.

Reviewer #2 (Remarks to the Author):

The authors present a concise and timely review on a relevant topic. While the review is short, this is an emerging area of research and all areas of relevant work was mentioned. The style of writing is clear, however, the authors must take more care to define their "graphene" terminology (ie what is a graphene-based material, what is GO, ect). Better explanations of data (for example, Figure 4, more details below) should be included to make the data more accessible to non-experts. To enhance readability, figures should be moved closer to their first mention in the text. Specific areas for improvement are outlined below. The choice of figures was quite good. Once these minor changes are made, I believe this review will be acceptable for publication in *Communications Chemistry*.

p. 1, Abstract: The difference between graphene and graphene-based materials needs to be made clear in the abstract. Here, they are used interchangeably, which is incorrect. The title also then needs to be adjusted accordingly.

p. 2, Introduction: The phrase "graphene-based materials" is used repeatedly, thus it may be easier to define an acronym for it (ie GBMs). Make sure to clearly define what class of graphene this abbreviation refers to.

p 2, line 3, Introduction: "cell response as reviewed in 1,2." must be changed to "... cell response as previously reviewed.1,2" In addition, citations typically appear after punctuation. Please check the full text change and change every other incorrect order of punctuation

p.2 Introduction: The intalic of physicochemical seem random. Either always or never italicize it.

p. 2, paragraph 2, Introduction: "confirmation/(s)" needs to be replaced with "conformation(s)"

p. 4 "Graphene substrate-serum protein interaction"

Please provide a more thorough description of GO. "a graphene derivative with oxygen incorporation", along with "negatively charged oxygenated functional groups" does not provide a very clear picture of the molecular structure of GO.

p. 5, [middle of the page] Please define the difference between pRGO and RGO

p. 11, bottom: "Their study demonstrated that GO promoted adipogenic differentiation of stem cells compared to graphene as shown in Figure 3(a)." Figure 3a shows a qualitative image at best. Please provide another metric that quantitatively and clearly shows that GO promoted adipogenic differentiation. Otherwise, please change the text to read "Their study suggested that GO promoted adipogenic differentiation of stem cells compared to graphene as shown in Figure 3(a)."

p. 13, Figure 3b- There is a scaling mismatch with the graphene in comparison to the proteins. Please note this in the caption.

p. 14, Figure 4. The description and figure for Figure 4a is not clear. The caption reads "Fluorescence micrographs illustrating pre-osteoblasts proliferation on BSA adsorbed scaffolds chitosan (CS-BSA) and chitosan-GO (CS-GO-BSA) after 7 and 28 days. Cells were stained with Hoechst to highlight nuclei." However, I only see scaffold features and not cell nuclei. This is likely due to autofluorescence of the scaffold which makes interpretation difficult, but should be explained.

p. 18, Figure 6a and b. There is a scaling mismatch with the graphene in comparison to the proteins. Please note this in the caption.

Reviewer Comments

Reviewer 1

1. Kumar and Parekh reported on the review of “Controlling cell fate 1 with graphene physicochemical properties: a rich, multiscale challenge”. The authors covered the manuscript including a. highlighting initial molecular confirmation of proteins on different physicochemical graphene-based surfaces, b. protein conformational changes, c. influence on cell-substrate interaction, and d. the connection to cellular function. Already, a series of reviews on graphene based materials have been reported previous. Almost similar type of review entitled “Interactions of graphene with mammalian cells: Molecular mechanisms and biomedical insights” was reported in *Advanced Drug Delivery Reviews* 105 (2016) 145–162. Therefore, this review does not have any significant novelty. In addition to this, some issues are associated with the manuscript which are described below:

Response: *In this review, we have attempted to provide a comprehensive picture of graphene-cell interaction, starting with the importance of molecular conformation of proteins on different physicochemical graphene-based surfaces, protein structural changes, influence on cell-substrate interaction, and the connection to cellular function. As the reviewer mentioned, several reviews have highlighted importance of graphene in biomedical field, but we would argue that none provide the complete picture starting from the perspective of protein molecular conformation on graphene substrates and building to cell function. The review that the reviewer mentions “Interactions of graphene with mammalian cells: Molecular mechanisms and biomedical insights” is good review and does discuss biomolecular interaction, such as protein adsorption on graphene forming a corona, which will influence biocompatibility. However, it does not discuss or highlight any link between the physicochemical features of graphene based materials, molecular structure of proteins (either during adsorption or afterward), and link this with cell control. In addition, the review does not discuss the mechanism by which the protein corona increases biocompatibility and their co-relation with protein structure in detail. Most importantly, it does not provide any information on how proteins on a graphene substrate can influence cell response such as differentiation. Hence, we believe our review provides significant extension by discussing how protein molecular structure on different physicochemical graphene-based surfaces can be correlated to cellular function. We hope that this review will encourage more researchers to work in this burgeoning sub-field and provide quantitative experimental results with direct connection between the different layers (graphene-protein and protein-cell) that link cell function and surface physicochemical properties of graphene-based substrates.*

2. The introduction section should be more informative.

Response: *While we respect the reviewer’s opinion, we are unclear how exactly to change the introduction to make it more informative as the reviewer did not provide any details toward this end. This review is combination of three themes (i) how different physicochemical graphene-based surfaces influence protein interaction, (ii) how protein molecular structure and orientation changes on different graphene substrates, and (iii) how cells sense such molecular structural changes of proteins on graphene substrates that modulates their behavior in terms of function and differentiation. As such, our goal was to give concise and relevant information in introduction section, which encourages readers to appreciate three themes of the review and consider them intertwined while reading the review article. A detailed introduction providing more information*

on each theme will be very lengthy and not appropriate for intended message. We have revised our introduction according to the Editor recommendations.

3. As the synthesis of graphene or graphene based materials is very important, so it should be included in the review.

Response: *We would like to thank the reviewer for mentioning the importance of synthesis of graphene or graphene based materials. We have added a brief intro on synthesis of graphene and graphene derived particles in updated manuscript.*

4. The authors should give a scheme, which describes the overall idea of the review.

Response: *In the updated manuscript introduction section, we have highlighted the theme and overall review idea into three main sections: (i) molecular conformation of proteins on different GBMs, (ii) how this influences cell-substrate interaction, and (iii) the connection to cellular response remains elusive.*

5. The toxicity of graphene-based materials has a major issue in biomedical application. Therefore, the toxicity effect should describe elaborately.

Response: *We agree with the reviewer and have added a new section on “GBM cytocompatibility” providing details on cytotoxic concerns related to graphene in the biomedical field.*

6. The title describes that the authors have covered the interaction between the graphene and different types of cells but the authors mainly focused on tissue engineering related work. If they want to describe on tissue engineering, then the title should be changed accordingly.

Response: *As mentioned above we wanted to show how protein molecular structure changes on graphene substrates influence cell function, which is often neglected in the field of tissue engineering. We have revised the title to highlight the three themes of our review.*

Reviewer 2:

1. The authors present a concise and timely review on a relevant topic. While the review is short, this is an emerging area of research and all areas of relevant work was mentioned. The style of writing is clear, however, the authors must take more care to define their “graphene” terminology (ie what is a graphene-based material, what is GO, ect). Better explanations of data (for example, Figure 4, more details below) should be included to make the data more accessible to non-experts. To enhance readability, figures should be moved closer to their first mention in the text. Specific areas for improvement are outlined below. The choice of figures was quite good. Once these minor changes are made, I believe this review will be acceptable for publication in Communications Chemistry.

Response: *We thank the reviewer for their positive feedback on our review and appreciate their constructive criticism. As the reviewer pointed out, we have ensured that the terminology in the manuscript is clearly defined for graphene based materials, GO and other graphene derivatives.*

2. p. 1, Abstract: The difference between graphene and graphene-based materials needs to be made clear in the abstract. Here, they are used interchangeably, which is incorrect. The title also then needs to be adjusted accordingly.

Response: *We agree with the reviewer and have updated the abstract (now preface) and introduction clearly stating the difference between graphene and graphene based materials (like GO, RGO and their chemical derivatives). We also updated and simplified the title.*

3. Introduction: The phrase "graphene-based materials" is used repeatedly, thus it may be easier to define an acronym for it (ie GBMs). Make sure to clearly define what class of graphene this abbreviation refers to.

Response: *Thank you for the suggestion. In the introduction, we have defined the meaning of “graphene-based materials” and coined the acronym GBMs as suggested by the reviewer. In the manuscript we have used term like graphene-based materials, graphene-based substrate and graphene-based particles which are synonyms to each other.*

4. p 2, line 3, Introduction: “cell response as reviewed in 1,2.” must be changed to “... cell response as previously reviewed.1,2” In addition, citations typically appear after punctuation. Please check the full text change and change every other incorrect order of punctuation

Response: *We have made the change in introduction as suggested by the reviewer and updated the citation appearance in according to Communications Chemistry journal style.*

5. Introduction: The italic of physicochemical seem random. Either always or never italicize it.

Response: *We have removed italic style of physicochemical word in the manuscript and made it normal format throughout the manuscript.*

6. p. 2, paragraph 2, Introduction: "confirmation/(s)" needs to be replaced with "conformation(s)"

Response: *We apologize for typo; the word “confirmation” has been corrected to “conformation” in the updated manuscript*

7. p. 4 “Graphene substrate-serum protein interaction” Please provide a more thorough description of GO. "a graphene derivative with oxygen incorporation", along with “negatively charged oxygenated functional groups” does not provide a very clear picture of the molecular structure of GO.

Response: *We have provided the definition of GO in introduction and also updated the details of GO in page 4 to provide a better understanding of GO molecular structure*

8. p. 5, [middle of the page] Please define the difference between pRGO and RGO

Response: *We have updated the difference between in pRGO and RGO in terms of difference in the amount of oxygenated functional group retention on the surface.*

9. p. 11, bottom: “Their study demonstrated that GO promoted adipogenic differentiation of stem cells compared to graphene as shown in Figure 3(a).” Figure 3a shows a qualitative image at best. Please provide another metric that quantitatively and clearly shows that GO promoted adipogenic differentiation. Otherwise, please change the text to read “Their study suggested that GO promoted adipogenic differentiation of stem cells compared to graphene as shown in Figure 3(a).”

Response: *We added quantification data in figure 3a as panel 3b, providing information demonstrating GO promoted significant adipogenic differentiation of stem cells compared to graphene. We have updated the figure accordingly in the manuscript.*

10. p. 13, Figure 3b- There is a scaling mismatch with the graphene in comparison to the proteins. Please note this in the caption.

Response: *The reviewer observation is correct, and we agree that the schematic lacks accurate scaling. As this figure is from a 3rd party research work, we have added the note in the caption stating “Note schematics of molecular substrate of protein (insulin), graphene and GO are not scaled proportionally”.*

11. p. 14, Figure 4. The description and figure for Figure 4a is not clear. The caption reads “Fluorescence micrographs illustrating pre-osteoblasts proliferation on BSA adsorbed scaffolds chitosan (CS-BSA) and chitosan-GO (CS-GO-BSA) after 7 and 28 days. Cells were stained with Hoechst to highlight nuclei.” However, I only see scaffold features and not cell nuclei. This is likely due to autofluorescence of the scaffold which makes interpretation difficult, but should be explained.

Response: *We apologize for the confusion in our wording. The contrast due to autofluorescence of the scaffold makes it difficult to see stained nuclei, which appear as, diffuse spots. We have added a note in the caption to inform the reader.*

12. p. 18, Figure 6a and b. There is a scaling mismatch with the graphene in comparison to the proteins. Please note this in the caption.

Response: *Similar to the comment on Figure 3, we agree that schematics presented in figure 6 are in proportionally scaled. We added this note in caption saying that “schematics of protein, GO and graphene are not scaled proportionally”.*